# The Effectiveness of Combining Botulinum Toxin Type A and Therapeutic Exercise in Treating Spasticity in a Patient with Complicated Stiff-Person Syndrome: A Case Report

**DOI:** 10.3390/diseases12060128

**Published:** 2024-06-17

**Authors:** Riccardo Marvulli, Maria Vittoria Raele, Mariagrazia Riccardi, Giacomo Farì, Maurizio Ranieri, Marisa Megna

**Affiliations:** 1Department of Translational Biomedicine and Neuroscience (DiBraiN), Aldo Moro University, G. Cesare Place 11, 70125 Bari, Italy; maryvi.92@hotmail.it (M.V.R.); mariagrazia.riccardi7@gmail.com (M.R.); maurizio.ranieri@uniba.it (M.R.); marisa.megna@uniba.it (M.M.); 2Department of Biological and Environmental Science and Technologies (Di.S.Te.B.A.), University of Salento, 73100 Lecce, Italy; giacomo.fari@unisalento.it

**Keywords:** stiff-man syndrome, stiff-person syndrome, botulinum toxin type A, therapeutic exercise

## Abstract

Stiff-person syndrome is rare and disabling autoimmune condition that most frequently affects women, with no real predisposition by race. Diagnosis is often arduous, which is why patients concomitantly suffer from anxiety and depression. To date, drug therapy is based on the use of benzodiazepines, barbiturates, and baclofen. Refractory cases are treated with intravenous immunoglobulin, plasmapheresis, B lymphocyte depletion with rituximab, and even the implantation of intrathecal baclofen devices. Botulinum toxin injection is frequently used, even if it still has an unclear role in the literature. Our case report aims to demonstrate the efficacy of a combined treatment of botulinum toxin and therapeutic exercise in a 65-year-old patient with biceps brachii muscle hypertonia and diffuse spasms of the axial musculature, using rating scales such as the Numeric Rating Scale (NRS) and Modified Ashworth Scale (MAS), joint range of motion (ROM) measurement, and muscle dynamic stiffness mensuration, which is performed by using the MyotonPro^®^. All the assessments were conducted at the first evaluation (T0), soon after the combined treatment with botulin toxin and therapeutic exercise (T1), three months (T2), six months (T3), and eight months after the botulinum toxin injection (T4). The patient demonstrated benefits for more than 6 months with no side effects. The combined therapy of botulinum toxin and therapeutic exercise had an excellent result in our patient.

## 1. Introduction

Stiff-person syndrome (SPS) or stiff-man syndrome (SMS), also known as Moersch–Woltman Syndrome from the men who first described it in 1956, is a rare autoimmune disorder characterized by a hyperactivation of the central nervous system (CNS) [1], leading to continuous, painful stimulus-triggered muscle spasms and rigidity [2]. Its estimated prevalence is 1–2 per million, with females being affected twice as often as males and without a clear predisposition for any race [3]. Usually, the anatomical parts affected by the pathology are the axial musculature, with the abdominal and paravertebral muscles involved as well as the proximal limbs, and it can, over time, lead to major skeletal deformities [4], causing disabilities with a significant psychological and social impact in the affected patients and forcing them to have frequent absences from work, which creates a huge impact on public spending [5]. Frequently, this condition is associated with anxiety and depression [6], which is why affected patients are usually directed to psychiatric physicians due to a failure in resolving the algic symptoms [7]. Based on clinical presentation, SPS can be classified into classic SPS and its variants, including stiff-limb syndrome (SLS) if it involves the limbs only [8] and progressive encephalomyelitis with rigidity and myoclonus (PERM) [9]. In both the classic and variant forms, autoantibodies against glutamic acid decarboxylase (anti-GAD), and specifically the GAD65 isoform, have been found in the cerebrospinal fluid of 60–80% of affected patients, confirming the hypothesis of the autoimmune basis of the disease [10]. A paraneoplastic form of SPS is recognized in about 5% of cases, which is still based on an autoimmune rationale but associated with a different profile of autoantibodies [11,12] and mainly with an involvement of the muscles of the neck and the upper extremities [2]. It appears that this form precedes the onset and diagnosis of cancer disorders, among which breast cancer, lung cancer, and lymphomas result the most frequent. Finally, there is an idiopathic form, which is still thought to be autoimmune in nature even though antibodies have not been discovered yet [13].

Since the diagnosis of this condition is always very arduous and is often identified as something less specific, some researchers have established specific criteria, called Dalakas criteria, that, when present, should at least make doctors consider the possibility of this syndrome. These criteria are the following: rigidity of axial and proximal limb muscles, painful muscle spasms triggered by noisy, bright stimuli or strong emotional states, positive response to diazepam, continuous muscle activity on electromyographic examination, and exclusion of other neurological pathologies [14]. Unfortunately, to date, this condition often remains undiagnosed and, consequently, not adequately treated [15]. Moreover, it now seems evident in the literature that while the classic form rapidly responds to drug therapies and has an excellent prognosis, the SLS, PERM, paraneoplastic, and chronic cases, which may present with predominantly brainstem involvement including generalized myoclonus (the so-called “jerking stiff-man syndrome”), are much more difficult to manage. [16].

Although the functions of different types of treatment have been now clarified, each affected patient should be carefully evaluated, and each should be given a specific, individualized therapy, either provided through a single drug or the combination of different medications. In general terms, the therapy with benzodiazepines, barbiturates, and baclofen per os is considered optimal in early stages [17]. However, side effects may preclude adequate dosing [18,19], and special attention should be paid to elderly patients and those with respiratory problems [20]. Sometimes, to lower the dosage of the just-mentioned drugs, they might be combined with Levetiracetam, Pregabalin, and even with Propofol, which has a big impact on the central nervous system [21]. Refractory cases find an advantage in the use of intravenous immunoglobulin (IVIG), plasmapheresis [22], and B lymphocyte depletion with rituximab, an anti-CD20 monoclonal antibody [23], although these can have even more serious side effects [24]. Intrathecal baclofen (ITB) can be used as a life-saving therapy for severe SPS, characterized by untreatable spasticity [25]. The safety and usefulness of botulinum toxin injection is already validated in the literature [26,27], although it still not widely adopted in such a disorder. Because of its well-known low side effects, its ease of administration, and its great effects in patients with spasticity due to other conditions for which botulinum toxin injections are already widely used, the aim of our case report is to demonstrate how a combined treatment with botulinum toxin type A injections into specific muscles in association with specific therapeutic exercise leads to satisfactory results, even in complicated patients.

## 2. Case Report

A 65-year-old Caucasian man diagnosed with SPS since 2021 required medical attention for the worsening of symptoms, specifically stiffness of the trunk and paravertebral musculature and the proximal part of the upper and lower limbs, as well as painful widespread muscle spasms mainly affecting the chest that had caused the patient to fall and no longer be able to stand up independently. The patient presented to the emergency room of the Bari General Hospital for the first time in 2021 for intense low-back pain refractory to painkiller intake, associated dyspnea, and clearly visible spasms of the axial musculature. He was admitted to the Neurology Unit for appropriate investigation and treatment. In both the serum and cerebrospinal fluid, the presence of anti-GAD-type antibodies was detected. Since the muscle spasms were visible, no electromyographic examination was performed, and a diagnosis of Stiff-Person Syndrome was made according to the above-mentioned Dalakas criteria. Moreover, according to what has been reported in the study of Newsome S.D. and Johnson T. in 2022, he has been classified as a “classic phenotype” of “definitive stiff-person syndrome spectrum disorders (SPSD)”, since he met all the major criteria for definitive diagnosis (clinical presentation, including typical body regions involved like torso and lower extremities > upper extremities; hallmark triggers for spasms/increased rigidity that, in our case, was an emotional stressor; hallmark exam findings: hyperlordosis, rigidity of torso and extremities, paravertebral spasms, spasticity in extremities, and gait and hyperreflexia; presence of serum autoantibodies to GAD65; exclusion of alternative diagnoses and no better explanation for syndrome) and two out of four minor criteria (presence of CSF autoantibodies to GAD65; robust response to muscle relaxers early on, which was diazepam for our case report) [28]. Having already been treated with diazepam since the diagnosis, he received a five-day treatment with IVIG, specifically from 22 July to 26 July 2022, resulting in a complete relief of symptomatology. In June 2023, he again required medical attention for the previously mentioned symptoms and was therefore newly hospitalized in the Neurology Unit of Bari General Hospital. After a careful medical evaluation and a series of blood tests, he was again treated with a five-day cycle of IVIG. Following the resolution of painful symptoms and muscle spasm reduction, he was discharged with a referral for a Physical Medicine and Rehabilitation evaluation due to his biceps brachialis (BB) rigidity that impaired his ability to perform some normal activities of daily living (ADLs) (e.g., brushing his teeth and his hair, eating independently, grasping objects, and washing himself with his right extremity). For this reason, in September 2023, the patient presented again to the Bari General Hospital and specifically to the toxin clinic in the Department of Physical Medicine and Rehabilitation.

At the first assessment (T0), soon after the combined treatment with 240 units (UI) of botulin toxin type A (Dysport^®^ https://www.dysportusa.com/) and therapeutic exercise (T1), three months (T2), six months (T3), and eight months after the botulinum toxin type A (Dysport^®^) injection (T4), the patient underwent the following: a medical examination with a joint range-of-motion (ROM) evaluation, rating scales to quantify the subjective perception of pain and functional limitation, and the BB dynamic stiffness mensuration, performed by using the MyotonPro^®^. Specifically, the rating scales were the following: the Numeric Rating Scale (NRS), one of the pain rating scales used to measure intensity of various symptoms, goes from 0, which indicates no pain, to 10, which represents the worst pain ever experienced; the second rating scale was the Modified Ashworth Scale (MAS), which consists in the most universally accepted clinical tool used to measure the increase of muscle tone (Table 1). The BB dynamic stiffness value was obtained using the MyotonPRO. Specifically, the instrument is placed with a uniform pretension (0.18 Newtons) to the undercut tissues in a perpendicular way with respect to the muscle, and it gives a 15-millisecond mechanical strike at a predefined force of 0.4 Newtons, followed by a rapid discharge. This whole procedure results in the onset of muscle oscillations that are recorded by the machine. Using this equipment, the biomechanical feature of the BB is calculated as a numerical value considering the vibration reduced by the muscle^.^ The accurateness of this measurement is well-documented in the literature [29].

At T0, the patient presented severe nuchal rigidity with impaired head flexion–extension and laterality movements; the right upper limb was positioned in 110° flexion with difficult passive mobilization (MAS = 2) and, specifically, a passive extension deficit of 50°. The BB dynamic stiffness parameter reported on the MyotonPRO was 410 N/m (Newton/meters). He also referred to very intense diffuse pain with a significant functional limitation (NRS: 10/10). On our recommendation, the patient started a cycle of therapeutic exercises, supervised by a physiotherapist at an affiliated center, for a total of four times a week, with the aim of modulating diffuse muscle spasms (at the paravertebral and thoracic levels) and right BB hypertonia. The exercises consisted of passive mobilization with the help of the physical therapist, active assisted mobilization of the right arm in flexion–extension and abduction–adduction, and mobilization of the neck in flexion–extension and rotation to the left and right, with the stretching exercises being maintained for at least 30 s each, as well as postural exercises that can strengthen the cervical and paravertebral muscles, including the trapezius muscles, with the aim of facilitating the sitting position and counteracting painful spasms. Great importance was also given to breathing exercises, both to manage axial muscle rigidity, which may impair breathing, and also because the patient we evaluated had been on diazepam therapy for a long time.

Moreover, to promote the reduction of BB muscle tone, focal ultrasound-guided treatment with botulinum toxin type A (Dysport^®^) was performed with a dosage of 240 UI.

Soon after the combined treatment with botulinum toxin type A and therapeutic exercise and exactly at the end of the physiotherapy treatment, which precisely was a month and a half after the first injection (T1), the patient underwent medical evaluation and rating scales again. He reported a marked improvement in algic symptoms with an NRS of 2/10. The spasms in the cervical and thoracic paravertebral muscles were significantly reduced, and this allowed the patient to improve both the posture of his head and neck as well as thoracic breathing. The right upper limb was flexed 30° with possible passive mobilization over the entire joint ROM as per an MAS 1+ hypertonia, with a passive extension deficit of 5°. The BB dynamic stiffness parameter reported on the MyotonPRO was 223 N/m. On this occasion, the patient was advised to continue with the therapeutic exercise at the same frequency as during the sessions and with the indication of targeted exercises for the stabilization of the obtained results.

To verify that the results could be long lasting and thus quantify their effects over time, the patients was also evaluated three months after the botulinum toxin type A (Dysport^®^) injection (T2). At this time, muscle spasms were not present and pain was slightly worse than at T1, although the patient considered it as a completely bearable discomfort with an NRS of 3/10. The right upper limb was flexed 30° with possible passive mobilization over the entire joint ROM as per an MAS 1+ hypertonia, with a passive extension deficit of 10°. The BB dynamic stiffness parameter reported on the MyotonPRO was 278 N/m. Also on this occasion, the patient was recommended to proceed with physiotherapy treatment over time.

A further evaluation was conducted on the patient at six months after the botulinum toxin type A (Dysport^®^) treatment, with the awareness that the toxin may have somehow lost its effect over time (T3). Fortunately, at that time, the patient reported an NRS of 2/10, and the right upper limb flexed 30° with the possibility of passive mobilization almost over the entire joint ROM as per an MAS 1+, with a passive extension deficit of 25°, thus demonstrating the resolution of the problem and lasting effects from the combined treatment of botulinum toxin type A (Dysport^®^) and therapeutic exercise over time. At this time, the BB dynamic stiffness parameter reported on the MyotonPRO was 289 N/m, supporting the idea that the situation had remained almost unchanged from the previous examination.

Eight months after the treatment with botulinum toxin type A (Dysport^®^) and therapeutic exercise (T4), a worsening of pain symptoms (NRS: 9/10) and spasticity (upper limb flexed at 110° with difficult passive mobilization as per an MAS = 3, with a passive extension deficit of 50°), compatible with a reduction in the effect of the botulinum toxin, was noted. Also, the BB dynamic stiffness parameter turned out to be 410 N/m. Despite this, sustained and continued therapeutic exercise over time improved the posture and reduced, almost solving, the frequency of painful spasms at the levels of paravertebral and neck musculature, in each case improving the patient’s overall posture. On this occasion, the same physician newly injected botulinum toxin type A (Dysport^®^) again into the patient’s BB with the same dosage (240 UI) and advised him to continue with the personalized therapeutic exercise with the same therapist (Figure 1, Figure 2, Figure 3 and Figure 4).

At the first appointment at the toxin clinic (T0), the signed written informed consent was collected, and the patient was asked to write down, in a provided diary, all possible side effects he might have had after the botulinum toxin injection. At T4, the diary was collected, and he reported no side effects, having benefited from the therapy, with a great improvement even in normal ADLs.

## 3. Discussion

As previously mentioned, SPS is due to CNS hyperactivation characterized by multiple muscle spasms and stiffness. Precisely for this reason, spasticity is a typical feature of these patients. Spasticity is a pathological condition consisting in involuntary and prolonged muscle contractions [30] that can be documented by the continuous muscle activity detected on the electromyographic examination [31]. Being caused by multiple neurological disorders such as cerebral palsy (CP) [32], stroke [33], and stiff-person syndrome itself, this condition has been extensively studied. Generically, spasticity treatments range from medical to surgical therapy, whatever the cause. The most implicated drugs are intramuscular botulinum toxin type A, oral baclofen, and supportive bracing. Surgical treatments, on the other hand, consist of intrathecal baclofen pump implantation and orthopedic procedures aimed at reducing deformities if painful and disabling, although these interventions, while not infrequent, tend to be used as a last resort. Among all these different approaches, botulinum toxin type A has proven to be a fundamental treatment for spasticity. Indeed, as early as 2000, a study conducted by Davis EC et al. demonstrated that botulinum toxin type A has resulted in a major advance in the field of movement disorders and in the management of spasticity due to its reversible yet long-lasting action, ease of administration, favorable safety, and adverse effect profile [34]. The use of botulinum toxin type A in patients affected by stiff-person syndrome is also well documented. A double-blind study conducted in 1995 demonstrated the efficacy of its injections in muscles suffering from continuous spasms and rigidity compared with saline solutions (placebo). Moreover, the same study described a botulinum toxin type A injection on one side that was able to decrease spasms in the contralateral part of the body as well, possibly because of the spread of the hematogenous toxin [35]. This means that in some patients, even with low dosages, it is also possible to have positive effects at other body sites besides the specific injected muscles but with a lower risk of side effects and good chances of a general improvement. The validity and extreme effectiveness of this treatment has also been proven for small muscles, such as the masseter and neck paraspinal muscles, in a 48-year-old male patient with uncontrolled SPS who was not responding to standard oral medications or the intrathecal baclofen pump [25]. Like all drug treatments, botulinum toxin injection is not free from side effects. Respiratory symptoms and infections were the most frequently registered adverse events in children with cerebral palsy, followed by asthenia [36]. Sentinel events, including four cases of death, were reported in the same study review. A study conducted in 2015 clustered the side effects into several groups: those inherent to the injected muscles: hypotonia, asthenia, hypokinesia, abasia, and dysstasia; those concerning the oropharynx: dysarthria, dysphagia, pharyngitis, feeding disorder, speech disorder, tongue paralysis, laryngospasm, vocal cord disorder, trismus, drooling, ageusia, stomatitis, aphagia, dry mouth, oral swelling, and so on; those involving the respiratory system, such as cough, dyspnea, pulmonary embolism, choking, and pneumonia; the eye-related side effects, among which the most common are blepharospasm and eyelid ptosis; the bowel- and bladder-related ones, such as constipation, nausea, vomiting, urinary incontinence, and bladder spasms; and finally, among the most common are those related to infections [37].

Obviously, it is important to emphasize the fact that the risk in the development of side effects may depend on the injected botulinum toxin dose. Thus, the aim of our case report was to attempt a combined treatment that could link an intensive rehabilitative exercise with slightly lower doses of botulinum toxin injected per muscle. In addition, the rationale of our study also lies in the fact that having patients follow a therapeutic exercise, to be started soon after botulinum toxin injection, and thus, the resulting mobilization, may also help the toxin in spreading, making some sites easier to reach. In fact, it is now well established in our outpatient experience that all patients with spasticity due to stroke, CP, or other acute events manage to have greater benefits and longer lasting effects from the toxin if they combine therapeutic exercise starting a few days after the injection.

In this regard, there is scientific evidence demonstrating the effectiveness of therapeutic exercise in SPS-affected patients. In 2002, it was already shown that therapeutic exercise could improve the quality of life (QoL) of 35-year-old woman with SPS with low back pain and stiffness [38]. In this specific case, the woman had been having worsening lower back pain for two years, which started insidiously and then became continuous over time. In addition to the pain, radiographically, the patient reported an accentuation of the lumbar curvature, leading to the sacrum being almost horizontal. Despite such an advanced and severe condition, it was shown how therapeutic exercise could positively affect this women’s QoL. Another case report published 4 years later brought to light how a physical therapist-supervised exercise program is completely necessary in this kind of patient, since they are affected by a chronic condition that should be controlled wherever possible. In fact, in addition to drug therapy, it is necessary for SPS patients to be able to initiate a therapeutic program that can be adjusted accordingly, given the frequently changing symptoms experienced by the patient [39]. Moreover, since stiffness and spasms interfere with the ability of these patients to fully mobilize the affected joints, leading to a higher risk of developing further complications, physical therapy and stretching could maintain joint mobility and therefore manage the disease over time [40]. Unfortunately, in this regard, the literature has not recorded actual specific treatment programs. What has emerged from the studies of some researchers is that stretching exercises, massages, relaxation exercises, exercises capable of improving the articular ROM, postural exercises, ultrasound, and hydrotherapy might be very helpful for SPS-affected patients. A study more than 10 years old highlighted the potential role of Pilates in treating stiffness [41]. Specifically, Clinical Pilates, so defined because it is designed purposely for the patient based on a careful assessment using a rating scale by a physician and physical therapist, is characterized by a set of exercises aimed to increase and protect the endurance, strength, and tone of the muscles in order to both lower the pain perception and improve the performance of the individual affected by stiff-man syndrome. In a precise case reported in the literature, the general principles on which Clinical Pilates was based on were concentration, breathing, which is essential since this function can be easily compromised in affected patients as mentioned above, focusing on the center, control, stability, and the activation of specific muscles [41].

Thus, the aim of our case report is to prompt researchers to reflect on the fact that since the therapeutic options for this type of disease are drugs, whose dosages are directly proportional to their side effects, it would be good to combine lower dosages and therapeutic exercise so as to improve patients’ QoL and lower the risk of their related side effects. Moreover, we believe it is improper to administer drug systemically (such as benzodiazepines and barbiturates), when in these patients, the residual problem only affects a few specific muscles (the right BB and the axial musculature in our specific case). In fact, it is both less harmful and even more useful to treat the involved muscle only, with therapeutic exercise where possible (as for the axial muscles of our case report) and with the association of botulinum toxin type A (Dysport^®^) injections when therapeutic exercise alone is not sufficient. Specifically, in a patient already treated with ad hoc therapies and with residual disease given by hypertonia of a muscle in the right upper limb, BB, the combination of 240 UI of botulinum toxin type A (Dysport^®^) and therapeutic exercise improved the patient’s quality of life for even longer than we hoped (at least 6 months). Indeed, based on what we observed in our specific case, the conclusion is that improving the subjective and objective parameters submitted by the patient by combining botulinum toxin type A injection and therapeutic exercise can facilitate the healing program by bettering the functional recovery of the upper limb in a shorter time, increasing the patient’s compliance, helping him to reintegrate into the common activities of daily living, and consequently, having an excellent effect on the patient’s state of mind.

Of course, this study is not free from limits. First, since it is a case report, we cannot claim with absolute certainty that the same effects will be seen in other patients even if affected by the same condition. In addition, as shown in our work and as is also clear in the literature, after an improvement given by the combination of the drug and therapeutic exercise, the effects tends to diminish over time, and therefore, this turns out to be an action that needs to be reintroduced to the patient after a period of time, which may vary from patient to patient but also for the same patient himself. Our intention is therefore to continue monitoring our 65-year-old Caucasian man to see if the time window of well-being remains constant rather than decreasing or even increasing. Moreover, we think it would be interesting, in future work, to carefully describe which kinds of exercises were performed in these patients so that we could make the study reproducible. For now, we think that having allowed the patient to conduct a similar, if not the same, lifestyle of a healthy person for 6 months could be considered as an impressive result.

## 4. Conclusions

SPS is a disabling condition that severely affects the suffering patients’ lives, forcing them to also abstain from normal activities of daily living. Although for some of them, per os therapy with benzodiazepines, barbiturates, and baclofen can be curative, there is a large portion of affected patients who do not respond and others who cannot intake these drugs at all, such as those with major respiratory problems. For our specific case, the combined treatment with botulinum toxin type A (Dysport^®^) supplemented with a targeted therapeutic exercise tailored to the patient, proposed by a qualified physiotherapist, resulted in a significant improvement in terms of both the subjective perception of pain (assessed using the NRS scale) and the functional limitation of the compromised segments (assessed using the measurement of joint ROMs, the MAS scale, and the BB dynamic stiffness by using the MyotonPro^®^), which remained constant up to six months after the botulinum toxin type A (Dysport^®^) injection. A future goal will be to monitor the patient over time and consider the disabling outcomes related to this rare syndrome. In addition, it would be desirable to propose the same type of treatment in a larger number of patients so that its effects can be better observed.

## Figures and Tables

**Figure 1 diseases-12-00128-f001:**
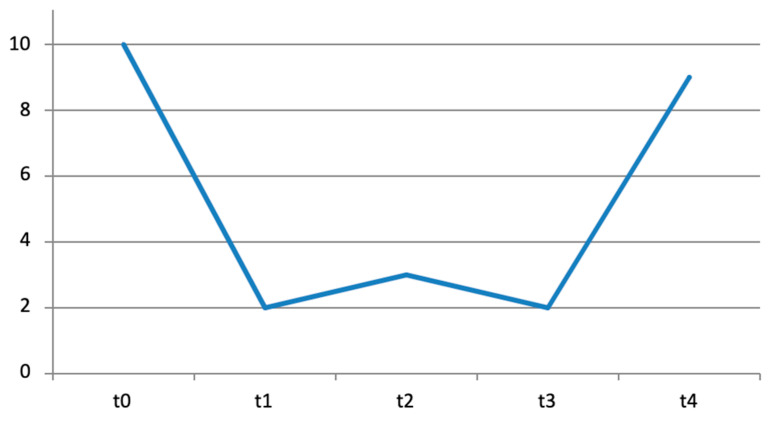
Pain trend quantified with the NRS scale at T0, T1, T2, T3, and T4.

**Figure 2 diseases-12-00128-f002:**
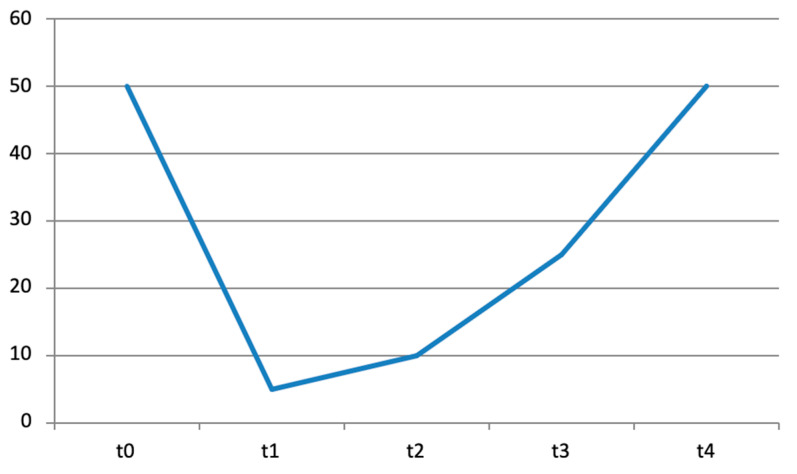
Trend over time of BB passive extension ROM deficit expressed in degrees at T0, T1, T2, T3, and T4.

**Figure 3 diseases-12-00128-f003:**
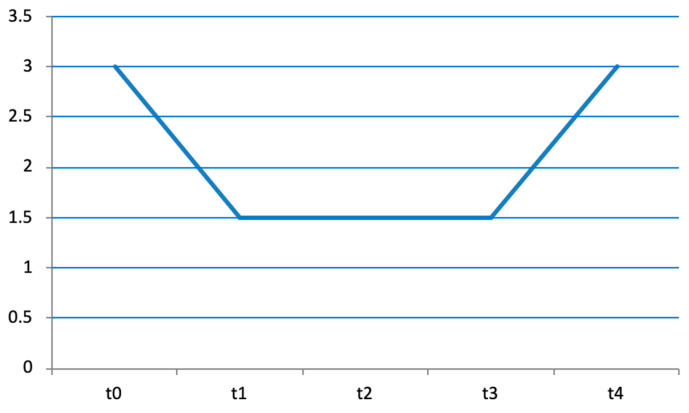
Trend over time of MAS scale, respectively, at T0, T1, T2, T3, and T4.

**Figure 4 diseases-12-00128-f004:**
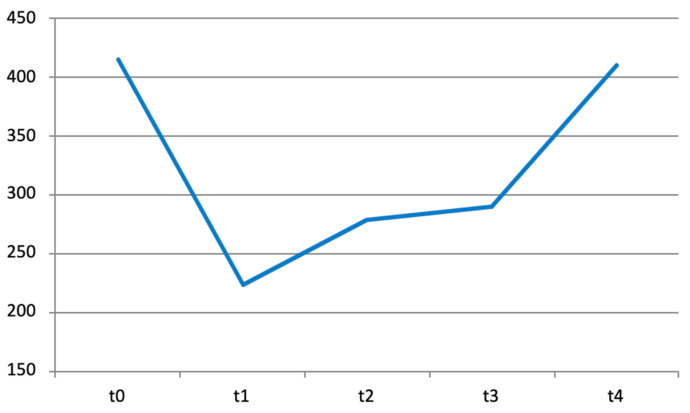
Trend over time of BB dynamic stiffness measured using the MyotonPRO^®^, respectively, at T0, T1, T2, T3, and T4.

**Table 1 diseases-12-00128-t001:** Modified Ashworth Scale (MAS).

0	No increase in muscle tone. Normal muscle tone.
1	Slight increased in tone, manifested by a catch and release or by minimal resistance at the end of the range of motion when the affected part is moved in flexion or extension.
1+	Slight increased in tone, manifested by a catch, followed by minimal resistance throughout the remainder (less than half) of the range of motion.
2	More marked increase in muscle tone through most of the range of motion, but affected parts are easily moved.
3	Considerable increase in muscle tone; passive movements difficult.
4	Affected part rigid in flexion or extension.

## Data Availability

The datasets used and/or analyzed during the current study will be made available upon reasonable request to the corresponding author, R.M.

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
