# Peer review of "The Effectiveness of Combining Botulinum Toxin Type A and Therapeutic Exercise in Treating Spasticity in a Patient with Complicated Stiff-Person Syndrome: A Case Report"

_diseases, 2024, doi:10.3390/diseases12060128_

Round 1

Reviewer 1 Report

Comments and Suggestions for Authors

The manuscript “The excellent results of combination of Botulinum toxin type A and therapeutic exercise in a patient with complicated Stiff man Syndrome: a case report” describes an exciting use of BoNTs to treat a rare autoimmune disorder. It is well-written and easy to read, making it worthy of publication after minor revisions. 

The authors should move some paragraphs of the section “Case report” to describe the events and the patient’s conditions chronologically. 

In addition, following there are some punctual comments:

-              Page 1, line 41. Please add a reference or comment to the statement, including the DALY or an estimation of the burden of this disease for the national health systems.

-              Page 3, lines 110 and 111. Please add the BoNT dosage.

-              Page 3, lines 117-125. The author should consider the opportunity to delete the Modified Ashworth Scale from the text and put the different values in a box (figure) or a table. This could improve the readability of the manuscript.

-              Page 5, lines 190 and 196. Please change the term “doctor” to “physician”.

Author Response

Point to point letter to reviewer 1.

Dear Reviewer,

We are proud of your positive comments. Thank you so much for them and for Your suggestions, which are great insights for us to improve our work.

Q1: The authors should move some paragraphs of the section “Case report” to describe the events and the patient’s conditions chronologically. 

A1: Thank you for this valuable suggestion. We modified the text of the “Case report” section by dividing it into paragraphs so that the evaluation at each appointment (from T0 to T4) is clearer.

Q2: Page 1, line 41. Please add a reference or comment to the statement, including the DALY or an estimation of the burden of this disease for the national health systems.

A2: Thank you for pointing this out and giving us the opportunity to clarify. Unfortunately, in the literature, the concept that health care spending due to this disease is impactful is clear from several studies, but none of them make the concept explicit enough, which is why we wrote the sentence without attaching any reference. It is fair, however, to emphasize this point. That is why we added a reference of a study that deals with all the complications related to this condition, ranging from psychiatric and cognitive aspects to gastric and respiratory disorders, to retinal problems and therefore, by analyzing all of them, better clarify the costs related to this condition, although always implicitly.

Q3: Page 3, lines 110 and 111. Please add the BoNT dosage.

A3: Thank you for this advice. Indeed, clarifying right away the dosage of botulinum toxin used makes the method clearer. We provided to add it in the text, as you suggested.

Q4: Page 3, lines 117-125. The author should consider the opportunity to delete the Modified Ashworth Scale from the text and put the different values in a box (figure) or a table. This could improve the readability of the manuscript.

A4: Thank you so much for your valuable advice. We added a table to make the main text easier to read. We believe this improved our work.

Q5: Page 5, lines 190 and 196. Please change the term “doctor” to “physician”.

A5: Thank you for the advice. We edited the text as requested and we think it is now improved.

Reviewer 2 Report

Comments and Suggestions for Authors

I thank the invitation to review the manuscript entitled “The excellent results of combination therapy of Botulinum toxin type A and therapeutic exercise in a patient with complicated Stiff man Syndrome: a case report”. The authors provided the clinical description of an elderly patient with Stiff man Syndrome treated with therapeutic exercise and Botulinum toxin type A. 

1.        I think it is of outstanding importance that authors must highlight that their attempt is mainly to treat spasticity in the context of Stiff-Person syndrome and not a treatment of the disease. I think this must be emphasized, for example, in the title of the manuscript (rewriting it in an understandable way to cite that it is directed to spasticity). 

2.        The best current terminology to be applied nowadays is Stiff-Person syndrome. 

3.        It is important to describe in the manuscript how was the patient’s diagnose of Stiff-Person syndrome performed. Did the authors perform neurophysiological studies? How was serological profile evaluated (i.e., serum or cerebrospinal fluid antibodies)? Did the patient fulfill Dalaka’s criteria? These aspects must be described in the manuscript for the reader. 

4.        There is no need in the manuscript to include the patient’s initials (A.P.). Authors should exclude these initials in all the manuscript content.  

5.        Have the authors any information about other previous therapies used by the patient (i.e., immunoglobulin; apheresis)? 

Author Response

Point to point letter to reviewer 2.

Dear Reviewer,

We are proud of your positive comments. Thank you so much for them and for Your suggestions, which are great insights for us to improve our work.

Q1: I think it is of outstanding importance that authors must highlight that their attempt is mainly to treat spasticity in the context of Stiff-Person syndrome and not a treatment of the disease. I think this must be emphasized, for example, in the title of the manuscript (rewriting it in an understandable way to cite that it is directed to spasticity). 

A1: Thank you for your advice which results fundamental in order to make the aim of the study easier and faster to understand. We modified the title as you suggested.

Q2: The best current terminology to be applied nowadays is Stiff-Person syndrome. 

A2: Thank you for pointing this out. We modified the terminology in the title and in the main text to make the work more in line with the current usage.

Q3: It is important to describe in the manuscript how was the patient’s diagnose of Stiff-Person syndrome performed. Did the authors perform neurophysiological studies? How was serological profile evaluated (i.e., serum or cerebrospinal fluid antibodies)? Did the patient fulfill Dalaka’s criteria? These aspects must be described in the manuscript for the reader. 

A3: Thank you for your valuable and crucial advice. Indeed, we realized that we explained the patient’s history very superficially and, therefore, following your pretious instructions, we modified the text by including more information regarding the diagnosis and Dalakas criteria.

Q4: There is no need in the manuscript to include the patient’s initials (A.P.). Authors should exclude these initials in all the manuscript content. 

A4: Thank you for your suggestion. This is extremely right, and we provided to exclude patient’s initials from all the main text.

Q5: Have the authors any information about other previous therapies used by the patient (i.e., immunoglobulin; apheresis)? 

A5: Thank you for the opportunity to better clarify this point. The answer is yes, we documented and are aware of all the therapies previously performed by the patient. As can be seen from the “case report” section, the patient has been using diazepam since diagnosis with benefit, and in the two flare-ups he underwent 5 days of immunoglobulin therapy, the first time with absolute success, the second time with persistence of spasticity in the mentioned muscle, treated by us with botulinum toxin and therapeutic exercise.

Reviewer 3 Report

Comments and Suggestions for Authors

The disease is very special, and the selected treatment methods also have certain new ideas, which still have certain clinical significance as a whole. 

But with only one case, not even a case series, is the treatment effect individualized and generalized to most patients? Its universality has not yet been known. Also, if this method is an old method, where is its innovation? If it is a new method, has it passed the ethical review? This problem needs to be further elaborated, otherwise the scientific rationality of the article will have a big problem. 

The pictures are just some data changes, there is no substantial evidence, and the degree of credibility is low. It seems that there are no actual cases that can make these pictures. In addition, it is not enough to reflect the changes in the course of the disease. Whether to consider supplementing more meaningful pictures as evidence. 

The overall content is long-winded, too long, and there are too many useless descriptions. Simplify the content of the article and reorganize the logical thinking. 

Author Response

Point to point letter to reviewer 3.

Dear Reviewer,

Thank you so much for Your suggestions, which are great insights for us to improve our work.

Q1: But with only one case, not even a case series, is the treatment effect individualized and generalized to most patients? Its universality has not yet been known. Also, if this method is an old method, where is its innovation? If it is a new method, has it passed the ethical review? This problem needs to be further elaborated, otherwise the scientific rationality of the article will have a big problem. 

A1: Thank you for giving us the opportunity to clarify these points.

Certainly, a major limitation of our study, as indeed that of all case reports, is that we tested this procedure exclusively on one patient and therefore could not be sure of the reproducibility of the therapy. In any case, our first goal is to follow the patient over time; the second one is to focus on patients with Stiff man syndrome in whom, after appropriate therapies, residual spasticity and treat them with botulinum toxin andtherapeutic exercise with the aim of further improving their quality of life and thus having larger case reports to expose.

The important thing is that this therapeutic proposal does not replace therapies for stiff man syndrome in any way but can be added in patients with residual spasticity to make their lives as normal as possible.

In addition, evaluation has been done not only with assessments such as ROM and MAS measurement, which, while very precise, are often interpretable by the physician, but also by MYOTONPRO. With the measurement of muscle stiffness thanks to this tool we have a standard and precise parameter to follow over time to assess actual improvement or worsening in a totally noninvasive way.

The other mainstay of our practice is therapeutic exercise, which in combination with botulinum toxin greatly enhances its effect, as well as being a moment of distraction and society integration for these kinds of patients, who often are affected by depression.

From our point of view,thanks to minimally invasive instruments (Myoton) and treatments (a single injection and therapeutic exercise), we were able to restore a patient with a complicated form of stiff man syndrome and associated disabling spasticity to a completely normal life for a long period of time (and we are still monitoring him after the second injection and he is still doing great).

Q2: The pictures are just some data changes, there is no substantial evidence, and the degree of credibility is low. It seems that there are no actual cases that can make these pictures. In addition, it is not enough to reflect the changes in the course of the disease. Whether to consider supplementing more meaningful pictures as evidence. 

A2: Thank you for your comment. We agree that changes in ROM degrees and changes in MAS scale might not be enough. This is precisely the reason why we supplemented with the measurement of stiffness of the biceps brachii muscle with MYOTON which provided us of an objective parameter of improvement following our proposed combined treatment.

Q3: The overall content is long-winded, too long, and there are too many useless descriptions. Simplify the content of the article and reorganize the logical thinking. 

A3: Thank you for your pretious advice. We provided to modify the main text following you suggestion were possible.

Round 2

Reviewer 2 Report

Comments and Suggestions for Authors

The authors have properly addressed and discussed the points suggested during review. As neurophysiological studies were not performed in the reported patient, my suggestion is that authors should consider the use of the Expanded Diagnostic Criteria for Stiff Person syndrome Spectrum Disorders, described by Newsome et al., 2022 (J Neuroimmunol 2022;369:577915), and include the category of their patient (baed on clinical and serological aspects). 

Author Response

Point to point letter to reviewer 2.

Dear Reviewer,
we would like to thank you for your positive comments and your valuable advice.

Q1: As neurophysiological studies were not performed in the reported patient, my suggestion is that authors should consider the use of the Expanded Diagnostic Criteria for Stiff Person syndrome Spectrum Disorders, described by Newsome et al., 2022 (J Neuroimmunol 2022;369:577915), and include the category of their patient (baed on clinical and serological aspects). 

A1: Thank you for your valuable suggestion. We read the article you recommended with much interest and added in the “case report” section the classification of our patient who, according to the scheme, falls into the “definitive diagnosis” and “classical phenotype” by fulfilling 5 major and 2 minor criteria. We will definitely use this classification in our future studies about this topic and believe it was a useful suggestion to improve our work.

Best regards

Reviewer 3 Report

Comments and Suggestions for Authors

bad revised

Comments on the Quality of English Language

bad revised

Author Response

Point to point letter to reviewer 3.

Dear Reviewer,
We are sorry you did not appreciate our work.

In any case, we had a native speaker proofread the case report to improve the English, and we tried to reduce as much as possible all the parts that, like you, we found long-winded and unhelpful.

We well understand your concerns about the use of a single case report, and we also know it is necessary to follow the patient over time and specially to try to reproduce the same therapy in other patients to prove its effectiveness. Anyway, we believe that the use of myotonPRO can be a non-invasive objective assessment of improvements over time that the literature currently lacks.

Moreover, since we did not perform the electromyographic examination, we tried to improve our work by classifying our patient according to the criteria already used in 2022 by Newsome SD and Johnson T.

Best regards.
